# Detection of *PTCH1* Copy-Number Variants in Mosaic Basal Cell Nevus Syndrome

**DOI:** 10.3390/biomedicines12020330

**Published:** 2024-01-31

**Authors:** Guido M. J. M. Roemen, Tom E. J. Theunissen, Ward W. J. Hoezen, Anja R. M. Steyls, Aimee D. C. Paulussen, Klara Mosterd, Elisa Rahikkala, Axel zur Hausen, Ernst Jan M. Speel, Michel van Geel

**Affiliations:** 1Department of Pathology, Maastricht University Medical Center, 6229 HX Maastricht, The Netherlands; to.theunissen@zuyderland.nl (T.E.J.T.);; 2GROW School for Oncology and Reproduction, Maastricht University, 6229 ER Maastricht, The Netherlands; aimee.paulussen@mumc.nl (A.D.C.P.);; 3Department of Dermatology, Maastricht University Medical Center, 6229 HX Maastricht, The Netherlands; 4Department of Clinical Genetics, Maastricht University Medical Center, 6229 HX Maastricht, The Netherlands; anja.steyls@mumc.nl; 5Research Unit of Clinical Medicine, Department of Clinical Genetics, Medical Research Center Oulu, Oulu University Hospital, University of Oulu, 90570 Oulu, Finland

**Keywords:** basal cell nevus syndrome (BCNS), *PTCH1*, MLPA, ddPCR, mosaicism, mosaic, CNV

## Abstract

Basal cell nevus syndrome (BCNS) is an inherited disorder characterized mainly by the development of basal cell carcinomas (BCCs) at an early age. BCNS is caused by heterozygous small-nucleotide variants (SNVs) and copy-number variants (CNVs) in the Patched1 (*PTCH1*) gene. Genetic diagnosis may be complicated in mosaic BCNS patients, as accurate SNV and CNV analysis requires high-sensitivity methods due to possible low variant allele frequencies. We compared test outcomes for *PTCH1* CNV detection using multiplex ligation-probe amplification (MLPA) and digital droplet PCR (ddPCR) with samples from a BCNS patient heterozygous for a *PTCH1* CNV duplication and the patient’s father, suspected to have a mosaic form of BCNS. ddPCR detected a significantly increased *PTCH1* copy-number ratio in the index patient’s blood, and the father’s blood and tissues, indicating that the father was postzygotic mosaic and the index patient inherited the CNV from him. MLPA only detected the *PTCH1* duplication in the index patient’s blood and in hair and saliva from the mosaic father. Our data indicate that ddPCR more accurately detects CNVs, even in low-grade mosaic BCNS patients, which may be missed by MLPA. In general, quantitative ddPCR can be of added value in the genetic diagnosis of mosaic BCNS patients and in estimating the recurrence risk for offspring.

## 1. Introduction

Basal cell nevus syndrome (BCNS), also known as Gorlin–Goltz syndrome, is characterized by the development of multiple nevoid basal cell carcinomas at an early age and is often accompanied by clinical features such as odontogenic keratocysts, palmar pits, congenital skeletal abnormalities, and an increased risk for the development of medulloblastomas [1]. BCNS is inherited in an autosomal dominant fashion and is, in a vast majority of cases, caused by a heterozygous loss of function variants in the tumor suppressor gene Patched 1 (*PTCH1*), which is part of the Sonic Hedgehog (SHH) signalling pathway. More rarely, BCNS has been reported to be caused by a loss of function mutation in *PTCH2* and *SUFU* [2,3], although the involvement of *PTCH2* has been disputed [4]. The SHH pathway drives cell proliferation and differentiation via the transcription factor Gli1 [5] and has a prominent role during embryonic development. Consequentially, BCNS patients often suffer from congenital growth abnormalities affecting the skull (e.g., macrocephaly), spine, and ribs. The development of BCCs, which is one of the major clinical criteria for the diagnosis of BCNS [1], is dependent on a somatic second-hit pathogenic variant in the other *PTCH1* allele, which is typically introduced via UV light exposure, causing multiple BCCs to appear at a young age. Additional somatic variants may be introduced in the SHH pathway, for example, in *Smoothened* (*SMO*), which may drive BCC development [6].

Although a clinical diagnosis of BCNS can be established based on the major and minor symptom criteria, a genetic diagnosis is important, not only to confirm the clinical diagnosis, but also to understand the disease segregation pattern and recurrence risk. The establishment of a genetic diagnosis for BCNS relies on the sequencing of the *PTCH1* gene in order to detect single-nucleotide variants (SNVs) and small indels but also requires accurate *PTCH1* copy-number analysis. Both large deletions and duplications in the *PTCH1* tumor suppressor gene have been reported to affect protein conformation and function, which results in PTCH1 inactivation [7,8]. Different methods exist for CNV analysis, including the conventional multiplex-ligation-probe (MLPA)-based methodology, targeted NGS-based approaches using, for example, smMIPs with unique molecular identifiers (UMIs), which help to detect CNVs in a more quantitative manner, and ddPCR, which allows absolute copy-number quantification [9,10,11,12,13]. For NGS-based CNV detection, a large reference dataset of control samples is needed for normalization, which may only be available after having analyzed enough samples. The choice of a specific CNV analysis methodology in a diagnostic setting is highly important, as the sensitivity and resolution eventually define the diagnostic yield of the genetic analysis [14]. This especially applies to CNV analysis in patients with a postzygotic mosaicism, who can be asymptomatic, and where the variant allele frequency in blood DNA can be very low due to spatiotemporal postzygotic occurrences in the embryonic tissues. Such CNVs might easily be missed when using conventional Sanger-sequencing-based approaches or MLPA due to a relatively low sensitivity [15]. Therefore, if available, the testing of different tissue types besides blood is an important diagnostic approach in suspected mosaic cases.

In this case study, we compared the test outcomes for *PTCH1* CNV detection using the more conventional multiplex ligation-probe (MLPA) methodology and digital droplet PCR (ddPCR) based on DNA isolated from the peripheral blood lymphocytes of a female BCNS patient (index patient), where MLPA detected a heterozygous CNV duplication. A smMIP-NGS approach for CNV detection was not possible since a large reference dataset was unavailable. The patient’s father, only presenting with multiple BCCs, was suspected to have a low-grade mosaic form of BCNS, and was therefore tested for CNVs based on multiple tissues, including blood, hair, and saliva. *PTCH1* CNV analysis in these two cases allowed us to compare the accuracy of the more conventional multiplex ligation-probe (MLPA) methodology with digital droplet PCR (ddPCR) for the detection of CNVs.

## 2. Materials and Methods

### 2.1. Patients and DNA Samples

At the Department of Clinical Genetics (MUMC+, Maastricht, The Netherlands), a request for genetic analysis was obtained for a female patient (index) from the Oulu University Hospital in Finland, who was clinically diagnosed with BCNS. Also, her father was analysed after the molecular diagnosis of BCNS was established. The 38-year-old patient met four major clinical criteria for BCNS, including multiple BCCs, palmoplantar pits, an odontogenic keratocyst in her jaw, and calcification of the falx cerebri [1]. She also met some minor criteria, including macrocephaly, kyphoscoliosis, an ovarian fibroma, and ocular abnormalities [1]. There were no family members with anamnestic symptoms of BCNS, but the patient’s father (68 years of age) had developed multiple BCCs at older age. To exclude BCNS, his blood was also analyzed after molecular diagnosis of BCNS was established in his daughter. The index patient’s mother, brother, and two children have no symptoms suggestive of BCNS.

Peripheral blood samples were collected from the index patient and her father, and genomic DNA was extracted using automated QIAsymphony device and Qiagen Qiasymphony DSP DNA Midi kit (Qiagen, Hilden, Germany) in Nordlab genetics laboratory, Oulu, Finland and sent to the MUMC+ for further genetic analysis. DNA isolation from the tissues (DNeasy Blood & Tissue Kit, Qiagen, Hilden, Germany) was performed in MUMC+ and used for genetic analysis.

Both patients provided written informed consent for genetic analysis, approved by the ethics committee of the Northern Ostrobothnia Hospital District (EETTMK: 45/2015 and amendment 2020).

### 2.2. PCR and Sanger Sequencing

*PTCH1* exons and flanking intronic regions were PCR-amplified using KAPA2G FastHotStart DNA polymerase combined with HotStart Buffer A (Merck, Darmstadt, Germany) using M13-labeled primers with a reaction concentration of 0.33 pmol/μL and 20 ng DNA input (isolated from blood lymphocytes). *PTCH1* primer sequences are specified in Appendix A, and PCR amplification was performed according to the manufacturer’s protocol. PCR products were purified using exo-sap treatment (Isogen life science, de Meern, The Netherlands). Sanger sequencing was performed using the BigDye Terminator Cycle Sequencing Kit (V1.1) (Thermofisher Scientific, Waltham, MA, USA) and the ABI 3730 automatic sequencer (Applied Biosystems, Foster City, CA, USA). Sequencing results were analyzed using the Mutation Surveyor software version 5.1 (Softgenetics, State College, PA, USA). *PTCH1* Sanger sequencing was performed only for the index patient.

### 2.3. MLPA

Multiplex Ligation-dependent Probe Amplification (MLPA) (MRC Holland, Amsterdam, The Netherlands) was performed using a DNA input of 200 ng, as measured by nanodrop, and diluted in MQ to a total volume of 5 μL. MLPA analysis was performed on DNA isolated from the (control) tissues. Also, a negative control condition was included. The sample hybridization mixture was prepared according to the manufacturer’s protocol, using SALSA MLPA Probe Mix P067-B2 *PTCH1* (MRC Holland, Amsterdam, The Netherlands). The MLPA kit contains 33 MLPA probes for 23 out of 25 exons of the *PTCH1* gene (no probes are included for exons 1a and 8, NCBI RefSeq: NM_000264.5), where amplification products sizes range from 142 to 454 nt. The probe mix contains ten reference probes located at different autosomal chromosome locations and nine quality control fragments with amplicons between 64 and 105 nt. The complete probe sequences in the P067 *PTCH1* mix can be found at the manufacturer’s website (https://www.mrcholland.com/ (accessed on 20 January 2024)). The MLPA probe hybridization mixture, MLPA ligation mixture, and PCR mixture were used according to the manufacturer’s protocol, as were the corresponding thermal cycle programs; 1 μL PCR product of each reaction (FAM-labeled) was used for capillary electrophoresis using the ABI 3730 automatic sequencer. Fragment sizes and signal intensities were quantified using the software tool GeneMarker V2.4.0 (Softgenetics, State College, PA, USA), and normalized to the reference probes (expected to have a normal copy number). A relative copy-number ratio was calculated with respect to the wild-type control DNA condition. All quality control fragments were checked to ensure they met the minimal quality criteria, ensuring sufficient DNA input (Q-fragments) and proper DNA denaturation (D-fragments) and avoiding sample mix up (X and Y fragments, no DNA control). A relative copy-number peak ratio of 0.75–1.25 was considered equivocal, whereas a ratio of >1.25 was interpreted as a copy-number gain, and <0.75 as a copy-number loss.

### 2.4. ddPCR

ddPCR-based CNV analysis of *PTCH1* was performed using the commercial BioRad PrimePCR copy-number assay for *PTCH1* (BioRad, Hercules, CA, USA), consisting of a duplex PCR with FAM-labeled *PTCH1* probe (channel 1) and HEX-labeled *EIF2C1* probe (channel 2). The 68nt *PTCH1* amplicon crosses an exon–intron junction and is detected by the FAM-labeled *PTCH1* probe. *PTCH1* copy-number quantification is corrected for the *EIF2C1* reference assay, which is based on a 69 nt amplicon detected using a HEX-labeled *EIF2C1* probe. Amplicon context sequences are indicated in Table 1.

As germline CNVs are most likely to involve less than 10 copies per cell, ddPCR was performed using ~20 ng DNA input of each tissue. ddPCR *PTCH1* CNV analysis was applied to the DNA samples, including a negative control. DNA fragmentation using the HaeIII restriction enzyme (New England Biolabs, Ipswich, MA, USA) was performed during the ddPCR assay in order to improve template accessibility, as recommended by the manufacturer (BioRad, Hercules, CA, USA). The reaction mixture was generated according to the manufacturer’s manual for ddPCR CNV analysis; 20 μL of the reaction mixture, containing the DNA, was added to the DG8 cartridge, as well as 70 μL of droplet generation oil for droplet generation in the QX200 Digital Droplet Generator. Thermal cycling was performed, followed by reading of the droplets using the QX200 Droplet Reader, all according to the manufacturer’s protocol (Biorad, Hercules, CA, USA). Data acquisition and analysis were performed using the BioRad QuantaSoft software v1.7.4, according to the manufacturer’s instructions. The *PTCH1* copy-number ratio was determined by calculating the ratio of the *PTCH1* target molecule concentration to the *EIF2C1* reference molecule concentration, multiplied by the number of copies of the reference molecule in the genome, which is 2. An average copy-number ratio was calculated based on five technical duplicates.

## 3. Results

### 3.1. Sanger Sequencing

DNA was isolated from the blood of the index patient, who was clinically diagnosed as having BCNS and tested for *PTCH1* germline variants via Sanger sequencing. No pathogenic *PTCH1* variant was identified that could explain the disease. However, noticeably, a heterozygous *PTCH1* polymorphism c.1504-51C>G in intron 10 (NCBI RefSeq: NM_000264.5) was detected in both the patient DNA and non-related healthy control DNA (Figure 1). In the healthy control DNA (Figure 1), the reference nucleotide (c.1504-51C) was detected with a variant allele frequency (VAF) of approximately 33%, and the polymorphism (c.1504-51G) with approximately 67%. This apparent imbalance probably does not reflect a realistic allelic imbalance and is most likely explained by the BigDye chemistry characteristics at this position. In the patient’s blood DNA (Figure 1), the VAF of the reference nucleotide and polymorphism were both ~50%, which suggested true *PTCH1* allelic imbalance in the index patient.

### 3.2. MLPA Analysis

For *PTCH1* genetic analysis, CNV quantification via MLPA analysis to detect exon deletions or duplications is part of the screening. Analysis of the index patient’s blood showed a copy-number gain (ratio > 1.25) for 9 out of 23 *PTCH1* probes in the region from exon 3 to 12 (NCBI RefSeq *PTCH1*: NM_000264.5). From these data, it is most likely that the CNV covers the complete exon 3–12 region (Appendix A) based on the distribution pattern of the probes. The index patient is heterozygous for a genomic (probably in-tandem) duplication c.(394+1_395-1)_(1728+1_1729-1)dup of exons 3 to 12 of the *PTCH1* gene (NM_000264.5). The average of the probe ratios in the amplified exon-3-to-exon-12 region was 1.445, which also suggested the presence of three *PTCH1* copies (~1.5 ratio), and thus a partial gene amplification involving exon 3 to 12 (Figure 2). Testing the blood of the index patient’s father indicated an identical MLPA probe distribution compared to the index, although not exceeding the >1.25 significant threshold level (average—1.11). Therefore, the father was suspected to be mosaic for the *PTCH1* genomic duplication. DNA from hair and saliva was isolated from the father to quantify the duplication CNV variant to estimate the variant distribution in different tissues. The *PTCH1* copy-number quantification was normalized to the reference probes, and a relative peak ratio was calculated with respect to the healthy wild-type control DNA (Appendix A). The MLPA analysis of the blood from the mosaic father did not, however, pass the threshold for *PTCH1* copy-number gain (Appendix A). The average relative copy-number ratio of the MLPA probes in the exon 3–12 region was 1.114 (Figure 2), showing that no *PTCH1* copy-number gain was detected based on the threshold value for copy-number gain. Although not considered a copy-number gain, this MLPA result would imply that the mutation is detected in ~23% [0.114/0.5 × 100] of the blood. CNV analysis of DNA isolated from the hair of the father showed an increased copy-number peak ratio for 8 out of 23 probes (Appendix A), with an average ratio of 1.370 with respect to the probes in the exon 3–12 region, indicating the presence of a *PTCH1* copy-number gain present in 74% [0.370/0.5 × 100] of the tissue. The CNV analysis of the saliva of the index’s father showed a relative copy-number gain for 7 out of 23 probes (Appendix A), with an average copy-number ratio of 1.297, indicating that a copy-number gain was present in 59%, yet was detected just above the cutoff threshold value (Figure 2).

### 3.3. ddPCR Analysis

We performed ddPCR analysis for absolute CNV quantification using the commercially available *PTCH1* CNV assay (BioRad), which consists of a single *PTCH1* target probe, located in the exon-10–intron-10 region, and a single *EIF2C1* reference probe for normalization, located in the exon-5–intron-5 region. Figure 3 shows the average *PTCH1* copy-number ratio calculated for each sample (Table 1). The analysis of the healthy control DNA indicated the high accuracy of the ddPCR analysis, as the average copy-number ratio was almost precisely 1.00 (0.99), with a small standard error between the technical duplicates. The latter confirmed a normal diploid *PTCH1* copy number. The ddPCR analysis of the index patient’s blood (DNA isolated from peripheral blood lymphocytes) indicated an average copy-number ratio of 1.51, which was suggestive of the presence of a germline *PTCH1* gain consisting of three copies, involving at least the exon-10–intron-10 junction region where the ddPCR probe was localized. Considering a germline CNV consisting of three copies, the detected average copy-number ratio almost precisely matched the theoretically expected value of 1.5, implying that the gain was detected in 100% of the blood lymphocytes. The analysis of the blood lymphocytes of the index’s father showed a slightly but significantly increased *PTCH1* copy number, where an average copy-number ratio of 1.13 was suggestive of a *PTCH1* mosaicism, present in ~26% [0.13/0.5 × 100] of the blood. Interestingly, the CNV was enriched in DNA isolated from the hair of the father, where a *PTCH1* gain was measured in approximately 92% of the tissue (average copy-number ratio of 1.46). In saliva, the *PTCH1* gain was measured in approximately 74% (average copy-number ratio of 1.37). The ddPCR data therefore show that the genetic mechanism underlying the BCNS phenotype in the index patient comprises a three-copy *PTCH1* gene gain that involves at least the exon-10–intron-10 junction, and which is inherited via the father, who has the CNV in a low-grade mosaic state.

## 4. Discussion

As illustrated in this case study, when performing targeted *PTCH1* genetic analysis in BCNS, non-quantitative detection methods such as Sanger sequencing might reveal indications for an allelic imbalance. Still, the testing for *PTCH1* copy-number variants in BCNS requires a sensitive and quantitative technique in order to establish reliable detection and an accurate diagnosis. A routinely used quantitative CNV detection method is MLPA, which is performed to exclude gross genomic deletion or duplication variants in DNA diagnostics for the *PTCH1* gene. In this study, we compared CNV analysis outcomes generated with MLPA and ddPCR methodology, based on a BCNS patient who was likely to harbor a germline *PTCH1* variant, as well as the mosaic father who presented with multiple BCCs only. Both ddPCR and MLPA analyses detected a significantly increased *PTCH1* copy number in the peripheral lymphocyte DNA of the BCNS patient (index) with an average duplication factor of 1.49 and an average relative copy-number ratio of 1.445, respectively (the average of the MLPA probes in the gained region). Although both outcomes are clearly indicative of a *PTCH1* copy-number increase, only the ddPCR outcome accurately reflected the presence of a germline CNV, in this case involving a *PTCH1* gene gain consisting of three copies. This also implies that MLPA detects the *PTCH1* gene gain in only 89% of the patient’s blood lymphocytes, which is approximately 11% lower than theoretically would be expected in a germline gene gain (copy-number ratio of 1.5), suggesting a less-accurate copy-number quantification. Moreover, the distribution of the MLPA probes revealed that *PTCH1* exons 3 to 12 were aberrant (with an increased copy-number factor (>1.25)), indicating a partial gene gain.

In contrast to the MLPA method, the ddPCR assay consists of only a single probe localized within the exon-10–intron-10 region of *PTCH1* and could therefore be considered less reliable, assessing only one genetic data point. However, in the peripheral blood lymphocytes of the index’s father, ddPCR analysis also detected an increased *PTCH1* copy number. The average duplication factor (1.13) indicated that the gene gain was present in ~26% of the blood lymphocytes, indicating that the *PTCH1* gene gain was inherited via the father, who was likely to harbor the gene defect as a low grade, postzygotic (gonadal) mosaicism. Interestingly, MLPA analysis was not able to detect an increased *PTCH1* copy number in the peripheral blood lymphocytes of the mosaic father based on the standard, validated cutoff threshold (>1.25) for copy-number gain (dept. of Clinical Genetics, MUMC+), where none of the 23 probes exceeded this threshold. Despite being considered as not significantly increased, the average relative copy-number ratio of 1.114 within the exon 3–12 region would suggest that the gain is present in ~23% of the father’s blood lymphocytes, with the latter showing a quantification difference of ~3% between ddPCR and MLPA.

The testing for CNVs in multiple tissues of a suspected mosaic patient is desirable, as it reduces the risk of missing the genetic defect due to an allele frequency that is below the technique’s detection sensitivity. The importance of such an approach is strengthened with regard to the mosaic patient, in whom the CNV was significantly enriched in DNA isolated from hair and saliva, and where both techniques were able to detect and confirm the *PTCH1* copy-number variant. Whereas ddPCR analysis showed that the *PTCH1* gain was present in hair cells (approximately 92%) and in saliva cells (74%), MLPA detected the gain in 74% of the hair cells and 59% of the saliva cells, indicating that a rather large variation exists between the ddPCR and MLPA measurement outcomes. Although CNV analysis in the gonadal cells of a mosaic patient can be of significant value for the estimation of the disease recurrence risk, no such analysis was performed in the spermatocytes of the mosaic father due to his declining this request and the irrelevance because of his non-reproductive age. This may, however, be relevant in other identified mosaic patients of reproductive age who do not yet have affected offspring.

Our data show that, although both techniques were able to detect the germline CNV in the index patient’s blood lymphocytes, MLPA can miss CNVs in the peripheral blood of mosaic patients, especially in low-grade postzygotic mosaicism. Thw=e limitation of MLPA in the establishment of a genetic diagnosis in mosaic patients was among others illustrated by a case study on clinically unaffected parents of patients with tuberous sclerosis complex and neurofibromatosis type 1, where it was shown that MLPA was also less sensitive than FISH or PCR for detecting large rearrangements [16,17]. In diagnostic settings, but also in most of the reported MLPA studies, a theoretical or arbitrary ratio range is commonly used, which is uniform for all probes in the mix, and defines the cut-off value with respect to the normal range (e.g., 0.75–1.25, 0.8–1.2, or 0.95–1.05) [16,18,19,20]. As illustrated in our case study, such a “safe” arbitrary ratio range could easily result in missing a CNV. Therefore, it has been proposed that we establish a reference cut-off range for each individual probe, and provide unequivocal scoring criteria for the more accurate identification of CNVs using MLPA [19,21,22]. Still, our data show that substantial differences exist with respect to the measured copy-number ratios in the different tissues between ddPCR and MLPA. Previously we showed the strength of ddPCR compared to RFLP in the detection of *PTCH1* mutations in low-grade mosaic BCNS [23]. When performing CNV analysis on DNA isolated from tissue with a high wild-type background, as is the case in mosaicism, absolute quantification methods such as ddPCR are even more valuable, where a high detection accuracy is accomplished that is not achieved when using relative CNV detection methods such as MLPA, QPCR, or array-based technologies [11,24,25]. Another example of the high sensitivity of ddPCR analysis was shown by detecting CNVs at an allele frequency of <1% in somatic skin mosaicisms [26].

A drawback to using ddPCR analysis for CNV detection in BCNS patients is the limited number of commercial CNV probes available for *PTCH1* [11]. In this study, a mutational analysis of the index patient’s blood revealed indications for an allelic imbalance covering the intron 10 region prior to MLPA and ddPCR analysis. Yet, with no such prior knowledge, the testing of BCNS patients in a routine diagnostic setting would require the development of *PTCH1* multiplex ddPCR CNV assays covering larger parts of *PTCH1*, thereby allowing more accurate mapping of the aberrant region within the gene [27]. The definition of a large reference dataset to normalize the smMIP-NGS data could overcome this limitation in CNV detection, and, in fact, cover the complete gene. One might consider cumulatively collecting diagnostic samples to feed this large reference set and eventually implementing this CNV analysis via NGS. However, this is time-consuming and therefore the design of custom-made probes for ddPCR may be faster and more sensitive in cases where a rare CNV variant is detected.

Although we concur that a study with only 2 cases and a small number of tissue samples is limiting, it has shown that accurate CNV analysis in BCNS patients is crucial to establish a reliable genetic diagnosis, which involves familial segregation testing. Furthermore, estimating the disease recurrence risk for offspring in mosaic patients to offer reliable genetic counseling in a prenatal diagnosis (PND) or preimplantation genetic test (PGT) context is pivotal. This especially applies to CNV testing in low-grade post-zygotic mosaicism patients that might not fulfill the diagnostic criteria for BCNS, and in which the genetic defect can be present in peripheral blood with a very low allele frequency. Particularly in these patients, ddPCR offers a more accurate method of copy-number quantification compared to conventional techniques such as MLPA.

## Figures and Tables

**Figure 1 biomedicines-12-00330-f001:**
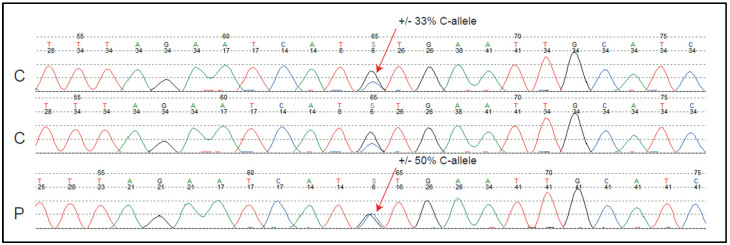
Sanger sequences of *PTCH1* intron 10 containing the heterozygous c.1504−51C>G polymorphism, which is present in both the blood DNA of healthy controls (C) and the index patient DNA (P). The polymorphism reveals a possible *PTCH1* allelic imbalance in the patient (33% C-allele represents two alleles and 50% C-allele represents three alleles).

**Figure 2 biomedicines-12-00330-f002:**
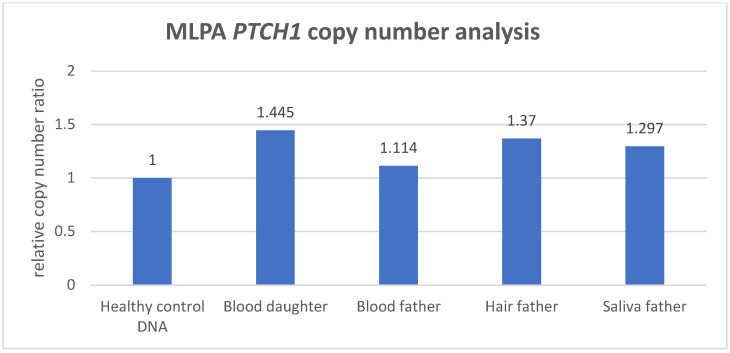
The relative *PTCH1* copy-number ratio based on the average ratio of the MLPA probes located from exon 3 to 12 (9 probes) with respect to the healthy control DNA. The CNV analysis of the index’s blood showed an increased copy-number (>1.25) ratio with respect to the probes in the region from exon 3 to 12, where a ratio of 1.445 was suggestive of a partial *PTCH1* gene amplification. The MLPA CNV analysis of the father’s blood did not indicate a copy-number gain (ratio—1.114), whereas MLPA did detect a copy-number gain in hair (ratio—1.370) and saliva (ratio—1.297).

**Figure 3 biomedicines-12-00330-f003:**
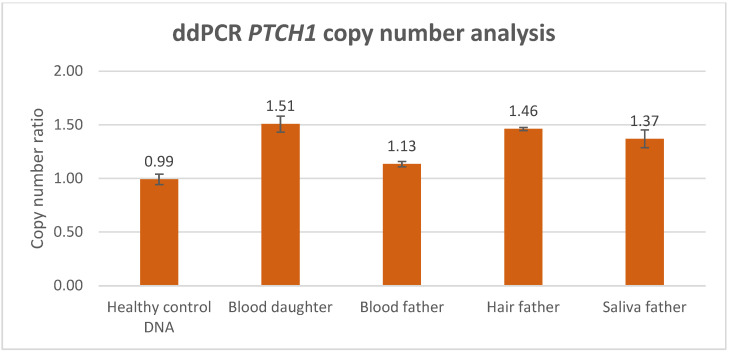
*PTCH1* copy-number ratio (*y*-axis) as measured via ddPCR analysis, based on DNA isolated from healthy control blood and the blood of the index patient and father, as well as from the hair and saliva of the father. Absolute *PTCH1* copy-number quantification was performed via normalization to the *EIF2C1* reference. Except for the healthy control DNA, all the tested samples showed an increase in the *PTCH1* copy number.

**Table 1 biomedicines-12-00330-t001:** ddPCR probe context sequences as indicated by the manufacturer (BioRad) and exonic–intronic location.

Probe	Amplicon Context Sequence	Probe Region
FAM-labeled *PTCH1* probe	CACAGCACACAGGAGGCTGGCTGGGCCAAGCCTGGG GGCCGGGTGGCATTTGTCAACGGACAGCAGATAAAT GGCTCCTTTAGTACCTGAGTTGTTGCAGCGTTAAAGG AAATTCCGATCAAT	exon 10–intron 10 (NM_000264.5)
HEX-labeled *EIF2C1* probe	GAGGGCTACTACCACCCGCTGGGGGGTGGGCGCGAG GTCTGGTTCGGCTTTCACCAGTCTGTGCGCCCTGCCAT GTGGAAGATGATGCTCAACATTGATGGTGAGTG GGGAGAGCTATGGAGC	exon 5–intron 5 (NM_012199.5)

## Data Availability

Data are available upon request.

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
