# Peer review of "Detection of PTCH1 Copy-Number Variants in Mosaic Basal Cell Nevus Syndrome"

_biomedicines, 2024, doi:10.3390/biomedicines12020330_

Round 1

Reviewer 1 Report

Comments and Suggestions for Authors

The authors have identified a patient with BCNS associated with a PTCH1 copy number variant (gain of PTCH1 exon 3 to 12). They also showed that her father carries the variant in mosaic form and developed BCCs at a later age. They compare MLPA and ddPCR methods and conclude that ddPCR is a more sensitive technique for the detection of mosaic variants.

Comments to the authors:

There are several calculations in the text comparing and quantifying the accuracy of each method for detection of mosaic variants, but it isn’t clear how many repeats were carried out for each method to demonstrate repeatability e.g.

Sections 2.2 & 3.1. For the PTCH1 polymorphism c.1504-51C>G in intron 10, how many times was each sample sequenced and were more control samples used that weren’t shown. If the samples were only sequenced once and only one unaffected sample was used, they experiments would need to be repeated for each sample at least 3 times for reproducibility and more control samples would be needed to ensure consistency and accuracy of the results.

Sections 2.3, 2.4, 3.2 & 3.3. How many unaffected control DNA samples were used for each assay (e.g. in my own laboratory, we routinely use 7 control DNAs for each MLPA experiment)? and how many times was each experiment repeated?

Section 3.2 It is a little confusing to say “Testing blood of the index patient’s father, indicated an identical MLPA probe distribution…” and then a couple of sentences later “MLPA analysis of the blood from the mosaic father did however not detect the PTCH1 duplication”. Perhaps this sentence could be removed or re-worded.

In the discussion, the authors compare the utility of MLPA and ddPCR for CNV detection e.g. ddPCR is more accurate, but has a limited range of probes within the gene. In the introduction, they also mention that smMIPs help to detect CNVs in a more quantitative manner, but require a large reference data set. Could the authors say a little more about their relative strengths and weaknesses e.g. would they recommend the use of each method for specific clinical diagnostic testing scenarios?

Minor comments:

The introduction should add a reference for SUFU-associated Gorlin syndrome (currently there only seem to be references for PTCH1 and PTCH2).

Gene names should be italicised throughout.

Comments on the Quality of English Language

The use of English is generally fine. There are just a few typos e.g. Section 2.2 “preformed”, Section 3.2 “and average”.

Section 2.1. What is the word “Anamnestic” used for?

Author Response

Dear reviewer,

Thank you very much for the valuable feedback on our manuscript. Your critical review has definitely improved the manuscript and I am grateful for the time and effort you  took.

In response to your comments, we addressed these comments point by point and indicated what adjustments we made to the manuscript.

The authors have identified a patient with BCNS associated with a PTCH1 copy number variant (gain of PTCH1 exon 3 to 12). They also showed that her father carries the variant in mosaic form and developed BCCs at a later age. They compare MLPA and ddPCR methods and conclude that ddPCR is a more sensitive technique for the detection of mosaic variants.

Comments to the authors:

There are several calculations in the text comparing and quantifying the accuracy of each method for detection of mosaic variants, but it isn’t clear how many repeats were carried out for each method to demonstrate repeatability e.g.

Regarding Sanger sequencing, at least a duplicate analysis is always performed in our practice when a new variant is found that is considered relevant.

The MLPA analysis is applied in a clinical diagnostic setting and for this a full validation has been performed in accordance with ISO15189 (2012), using both positive and negative (Reference) samples. In addition, per MLPA experiment, controls are always included (healthy individual and blank) as advised by the manufacturer. Thus, normalization is always performed against a reference established in the validation. Regarding MLPA, the blood of the index patient was analyzed three times, with the averages of each individual test resulting in the following values: 1.445 (as was used in the manuscript), 1.405 and 1.361. For the father, each MLPA analysis was performed only once. However, the index patient shows a relatively large variation from the expected value of 1.5, so in our opinion, further repetition of the analyses has little added value.

The ddPCR analysis was performed for each sample at least in 3-fold for both the index patient and the father. If desired, the raw data can be viewed.

Sections 2.2 & 3.1. For the PTCH1 polymorphism c.1504-51C>G in intron 10, how many times was each sample sequenced and were more control samples used that weren’t shown. If the samples were only sequenced once and only one unaffected sample was used, they experiments would need to be repeated for each sample at least 3 times for reproducibility and more control samples would be needed to ensure consistency and accuracy of the results.

The sample from the index patient was analyzed three times, with the SNP on PTCH1 c.1504-51C>G being well detected in both the forward and reverse sequence.

The SNP in question was detected in at least 7 other heterozygous individuals in our practice, all showing the same heterozygous pattern in both the forward (shown in the figure) and reverse sequences.

Sections 2.3, 2.4, 3.2 & 3.3. How many unaffected control DNA samples were used for each assay (e.g. in my own laboratory, we routinely use 7 control DNAs for each MLPA experiment)? and how many times was each experiment repeated?

Sanger: One control sample is always included per experiment and samples are also sequenced in both forward and reverse direction. In addition, a digital control (previously ran sample) is always loaded in the Mutation Surveyor software for sequence comparison.

MLPA: Please see our comment to note 1 for this. Additionally, all unaffected samples on which this MLPA analysis for PTCH1 was previously performed (>>100 samples) can be considered a reference control on which variation can be tested. Within the same experiment, as already mentioned, the manufacturer's recommended internal controls are included. This is part and parcel of a validated diagnostic test and thus also applies to this MLPA analysis. If a variant is seen, it is preferably confirmed on an independent DNA sample.

ddPCR analysis: In each experiment, a normal human control sample was included at least in triplicate.

Section 3.2 It is a little confusing to say “Testing blood of the index patient’s father, indicated an identical MLPA probe distribution…” and then a couple of sentences later “MLPA analysis of the blood from the mosaic father did however not detect the PTCH1 duplication”. Perhaps this sentence could be removed or re-worded.

Sentence is re-worded (line number 220).

In the discussion, the authors compare the utility of MLPA and ddPCR for CNV detection e.g. ddPCR is more accurate, but has a limited range of probes within the gene. In the introduction, they also mention that smMIPs help to detect CNVs in a more quantitative manner, but require a large reference data set. Could the authors say a little more about their relative strengths and weaknesses e.g. would they recommend the use of each method for specific clinical diagnostic testing scenarios?

See adaptations in the paper (line numbers 354-358).

Minor comments:

The introduction should add a reference for SUFU-associated Gorlin syndrome (currently there only seem to be references for PTCH1 and PTCH2).

Reference (Smith et al. J Clin Oncol. 2014 Dec 20;32(36):4155-61.) is added (line number 43).

Gene names should be italicised throughout.

Required adaptations are made throughout the manuscript.

 The use of English is generally fine. There are just a few typos e.g. Section 2.2 “preformed”, Section 3.2 “and average”.

Typos are corrected

Section 2.1. What is the word “Anamnestic” used for

Regarding history and circumstances relevant to the disease or condition.

We hope that the modifications made to the manuscript are acceptable to you and thank you again for considering this manuscript for publication in Biomedicines.

We look forward to your response.

Sincerely,

Guido Roemen

Manager, Dept. of pathology, Maastricht UMC+

P. Debyelaan 25 | 6229 HX Maastricht

E guido.roemen@mumc.nl | T +31(0)43-3874605

Reviewer 2 Report

Comments and Suggestions for Authors

I enjoyed reviewing this article entitled "Detection of PTCH1 copy-number variants in Mosaic Basal Cell Nevus Syndrome".

 Indeed, genetic diagnosis of basal cell nevus syndrome is crucial for these patients' clinical and therapeutic management and for calculating the disease recurrence risk for offspring. In mosaic forms, detecting genetic mutations can be tricky, and the authors describe a case study regarding genetic, molecular analysis in a patient with BCNS syndrome and her father, resulting in a post-zygotic mosaic form.

In particular, the authors aim to compare test outcomes of PTCH1  copy number variants (CNV) detection using multiplex ligation-probe amplification (MLPA) and digital droplet PCR (ddPCR). Results show that ddPCR is more accurate in detecting CNVs.

The article is globally well-written, especially the methods section. Therefore, some points need to be clarified: 

- The authors should provide the genetic tree diagram for the case study 

-Limitations of the study should be clearly stated

Author Response

Dear reviewer,

Thank you very much for the valuable feedback on our manuscript. Your critical review has definitely improved the manuscript and I am grateful for the time and effort you  took.

In response to your comments, we addressed these comments point by point and indicated what adjustments we made to the manuscript.

The article is globally well-written, especially the methods section. Therefore, some points need to be clarified: 

- The authors should provide the genetic tree diagram for the case study 

In section 2.1, the text briefly but completely presents the family composition and history, including individual symptoms (see line numbers 95-99). Therefore a genetic tree diagram is to our opinion not of added value.

- Limitations of the study should be clearly stated

Please see adaptations in the paper (line numbers 359 and 360).

We hope that the modifications made to the manuscript are acceptable to you and thank you again for considering this manuscript for publication in Biomedicines.

We look forward to your response.

Sincerely,

Guido Roemen

Manager, Dept. of pathology, Maastricht UMC+

P. Debyelaan 25 | 6229 HX Maastricht

E guido.roemen@mumc.nl | T +31(0)43-3874605

Round 2

Reviewer 1 Report

Comments and Suggestions for Authors

The authors have now addressed each of my comments and the manuscript is clearer.